Knowledge about cervical cancer and awareness about human papillomavirus vaccination among medical students in Jordan

Alsous Mervat M. mervat.alsous@yu.edu.jo 1
Ali Ahlam 2
Al-Azzam Sayer 3
Karasneh Reema 4
Amawi Haneen 1
1 Department of Clinical Pharmacy and Pharmacy Practice, Faculty of Pharmacy, Yarmouk University , Irbid , Irbid Governorate , Jordan
2 Medical Biology Centre, School of Medicine, Dentistry and Biomedical Sciences, Queen’s University Belfast , Belfast , United Kingdom
3 Department of Clinical Pharmacy, Faculty of Pharmacy, Jordan University of Science and Technology , Irbid , Irbid Governorate , Jordan
4 Department of Basic Medical Sciences, Faculty of Medicine, Yarmouk University , Irbid , Irbid Governorate , Jordan
Mitsouras Katherine
Electronic publication date: 2021 Jun 17
Publication date: 2021
Volume: 9
Electronic Location ID: e11611
Received 2021 Mar 12; Accepted 2021 May 24
Copyright: ©2021 Alsous et al.
Copyright year: 2021
Copyright holder: Alsous et al.
License: This is an open access article distributed under the terms of the Creative Commons Attribution License, which permits unrestricted use, distribution, reproduction and adaptation in any medium and for any purpose provided that it is properly attributed. For attribution, the original author(s), title, publication source (PeerJ) and either DOI or URL of the article must be cited.
License URL: https://creativecommons.org/licenses/by/4.0/

Keywords: Cervical cancer, Jordan, Knowledge, Medical students, Human Papillomavirus, Vaccine

Funding: The authors received no funding for this work.

==============================
Objectives

To assess the knowledge about cervical cancer and HPV infection and the awareness towards and perceived barriers of HPV vaccination amid medical students in Jordan.

Methods

The present study is a cross-sectional survey that was conducted for a period of three months in the College of Medicine at six different universities in Jordan. Third-year to sixth-year students from these medical colleges in Jordan were invited to participate in the study.

Results

There were 504 students that took part in the study with 42.3% being males and 57.7% females. The mean knowledge score of students in our survey was 21.4 ± 4.4 out of 34, which was categorized as a moderate level of knowledge regarding cervical cancer and HPV. Only 40.5% knew about the availability of the HPV vaccine in Jordan, and 65.9% accepted the idea that it is necessary to introduce the HPV vaccine for school girls in Jordan.

Conclusions

This study highlights that there is inadequate knowledge about cervical cancer and its screening among medical students in Jordan. Despite the limited awareness about the HPV vaccine among the study’s participants, there is a favorable opinion towards the introduction of the vaccine for school girls in Jordan. The data provide a benchmark on the level of knowledge about cervical cancer and awareness about HPV, which can be used to formulate an effective awareness program.

Introduction

Cervical cancer (CC) is the fourth most common cancer in females with 270,000 women dying from the disease annually; 90% of whom live in developing countries. In North Africa and the Middle East, CC affects 19,500 women per year, leading to 9,930 deaths annually. By 2035, these numbers are expected to double in this region unless effective public health interventions are introduced (World Health Organization, 2020).

CC, caused by human papillomavirus (HPV), is the major cause of years of life lost to cancer in the developing world. Since it affects women in their most productive years, CC has a disturbing effect on the well-being of families (World Health Organization, 2019). In Jordan, current estimates indicate that 104 women are diagnosed with cervical cancer every year and 61 die from the disease. Data on the HPV burden in the general population of Jordan is not yet available (ICO/IARC Information Centre on HPV and Cancer, 2018). However, in Western Asia, where Jordan is located, 72.4% of invasive cervical cancers are attributed to certain types of HPV (World Health Organization, 2020). Therefore, conducting comprehensive evaluations of HPV prevalence and examining the knowledge, attitudes, and practices toward HPV vaccination will provide a clear description of the situation in the region. Introducing a successful HPV vaccination program will directly reduce morbidity and mortality from HPV types, improve women‘s health, increase healthcare cost savings, and extend positive externalities on women’s immediate communities.

Amongst all known risk factors, persistent infection from high-risk HPV plays a considerable role in the pathogenesis of CC. The available screening tests for cervical cancer include conventional and liquid-based cytologic tests (i.e., Pap tests) while the most common test to detect HPV is to use a polymerase chain reaction (PCR) assay (Gargano et al., 2017). Almost all cases of CC are attributed to HPV, with subtypes 16 and 18 accounting for more than half of the CC cases worldwide (Schiffman & Castle, 2003; Trottier & Franco, 2005; Franco & Harper, 2005; Munoz et al., 2003; Lee & Chan, 2012). According to the International Agency for Research on Cancer (IARC), ten additional HPV types (31, 33, 35, 39, 45, 51, 52, 56, 58, and 59) are associated with cervical cancer as they have adequate evidence of carcinogenicity in humans. On the other hand, other HPV types, including types 6 and 11, can cause genital warts, benign or low-grade cervical cell changes, and recurrent respiratory papillomatosis (Gargano et al., 2017).

Successful achievements in basic and clinical research have expanded the possibilities of CC prevention by introducing HPV testing as part of the screening technology and, most importantly, by the production of efficacious prophylactic HPV vaccines (Franco & Harper, 2005). It is established that well-organized cervical screening programs or widespread good quality cytology can reduce CC incidence and mortality. Diagnostic screening programs for HPV lesions are generally available in the developed countries. However, all Middle East countries including Jordan do not have a national CC screening program due to the lack of public health policy, professional and general education, clinical settings, financial resources, and media awareness. It is noteworthy that most female cancer awareness campaigns in the Middle East are mainly focused on breast cancer (Jumaan et al., 2013).

The first vaccine against HPV for the primary prevention of CC was licensed in 2006 by the Food and Drug Administration (FDA) of the United States of America (USA). Currently, two prophylactic vaccines against HPV are registered: quadrivalent Gardasil (Merck, USA) and bivalent Cervarix (GlaxoSmithKline, Belgium). Both vaccines have good profiles for efficacy in preventing HPV infection and are well tolerated. Published studies have recognized a high efficacy of over 92% against precancerous lesions among women (World Health Organization, 2009). However, the uptake of these vaccines has been slow in the extended Middle East and North Africa regions due to several factors such as financial limits, weak infrastructure for adolescent vaccine delivery, lack of reliable data on the burden of HPV disease, and competition with high priority vaccines. Other barriers include religious and cultural sensitivities as the vaccines are given to prevent a sexually transmitted disease in young girls.

Several studies, mostly from developed countries, have demonstrated that the knowledge about HPV infection and the acceptability of HPV vaccines amongst health care professionals and the general public vary from low to high (Chawla, Chawla & Chaudhary, 2016; Warner et al., 2017; Pereira et al., 2019; Sherman et al., 2020). Recommendation of HPV immunization by physicians has been recognized as one the most significant factors in the individual’s willingness to receive the vaccine. Updated knowledge about the HPV vaccine and the elimination of any barriers to prescriptions among physicians are the main determinant factors (Riedesel et al., 2005; Pandey et al., 2012).

Physicians can play an imperative role in circulating knowledge about CC and the available preventive vaccines. Therefore, the awareness of physicians and medical students will greatly impact the success of CC prevention. The awareness programs should focus on family physicians and gynecologists depending on the known immunizer in each country. Targeted community education that responds to concerns about the HPV vaccine has proven to be effective (Leung et al., 2019). It is therefore important to enhance promotion, communication, and social mobilization strategies to increase awareness among decision makers and demand among the population. The strategy should include several companion organizations in the communities such as non-governmental organizations, women’s groups, religious leaders, and professional organizations.

The objectives of this study were to evaluate the knowledge about CC and HPV infection and awareness towards and perceived barriers of HPV vaccination among medical students in Jordan who have the essential role as health care providers to raise community awareness and to modify population behavior. This is the first time in Jordan that the assessment of their knowledge about CC and HPV is being done.

Methods

Study design and purpose

The present study is a cross sectional survey that was designed to investigate the knowledge about CC and awareness about the HPV vaccine among medical students in Jordan.

Study participants and ethical considerations

The protocol of the study was approved by the Scientific Research Committee at Yarmouk University. The study was conducted after obtaining ethical approval from the Institutional Review Board (IRB) of Jordan University of Science and Technology (JUST) and King Abdulla University Hospital (KAUH), Irbid, Jordan (13/128/2019).

The questionnaire was distributed utilizing an electronic format, through Google Forms. The link to the survey was shared with medical students in six universities in Jordan. The questionnaire was prefaced by a page explaining the nature and objectives of the study and the voluntary nature of participation with a consent statement if they would like to take part in the study. Participants who completed the questionnaire gave electronic informed consent by declaring their acceptance to fill out the questionnaire. The questionnaire was terminated automatically if participants declined to take part. This procedure was approved by the IRB committee.

The participants were assured that the outcomes of the research would not be used for routine appraisal of the participants. The individuals were requested to complete the questionnaire without textbooks or consulting materials.

The study was conducted for a period of three months in the College of Medicine at six different universities in Jordan. Third-year to sixth-year medical students from these medical colleges in Jordan were invited to participate in the study.

Study instruments

The study questionnaire was developed by authors after an extensive review of literature. A pre-validated questionnaire, consisting of items modified from questionnaires in other studies (Pandey et al., 2012; Aljuwaihel et al., 2013) was used. The questionnaire was reviewed by the authors and then subjected to pilot testing by 30 participants to ensure the clarity of the questions, which resulted in several minor amendments.

The final version of the questionnaire consisted of three parts. The first one was about the demographic information of participants which included age, gender, year of study, and average monthly income. The second part assessed the knowledge of participants about CC and HPV. Finally, the third part assessed the participants’ awareness and acceptance of HPV vaccination.

The knowledge of the students about CC was evaluated using 14 multiple choice questions with 34 statements related to disease etiology, risk factors, clinical features, and screening recommendations according to the World Health Organization (WHO). Each answer was scored as incorrect or correct. The respondent was given a zero point for each wrong answer and one point for each correct answer.

The total knowledge score was calculated for each participant out of 34. Participants were categorized to have poor, moderate, and good knowledge if their score was 0–17, 18–25, and 26–34, respectively.

Regarding respondents’ awareness about HPV vaccination, this part consisted of seven questions. Each question was scored out of two points. A poor awareness level was allocated to the medical students with a maximum 39% of awareness mean score (0–5 points), an average awareness to those with at a 40–69% of awareness mean (6-9 points), and a good awareness level to people with over 70% of awareness mean (10-14 points).

The last three questions in the questionnaire assessed the acceptance of HPV vaccination among medical students in Jordan, the perceived barriers of HPV vaccination, and their source of information about it.

Statistical analysis

Data was analyzed using SPSS software version 24. Descriptive data was expressed as frequencies and percentages. Chi square was used to analyze significant differences between categorical variables. Student’s t-tests were used to compare the means between two groups. Analysis of Variance (ANOVA) was used to compare the means between three or more groups. Analysis of Variance (ANOVA) was used to compare the means between three or more groups. The questionnaire’s internal consistency and reliability was assessed by measuring Cronbach’s alpha coefficient (α), values ≥ 0.70 were considered adequate. All p-values were two sided and any p-value of less than 0.05 was considered statistically significant.

Sample size calculation

Around 9,218 students are currently enrolled to study medicine at Jordanian universities (Goussous, 2016). The sample size was calculated utilizing the online Raosoft software sample size calculator. The minimum required sample size assuming a 95% confidence level, 50% recruitment rate, 5% margin of error, and a maximal sample size of 6,000 students would be 362 participants.

Results

Demographic characteristics

The number of total responses was 508, with 4 students disagreeing to take part in the study and 504 of the students completing the questionnaire. Recruited students were from all six medical colleges in Jordan including Yarmouk University (33.3%), Jordan University of Science and Technology (25.0%), University of Jordan (16.1%), Hashemite University (13.5%), Mutah University (6.3%), and Albalqa University (5.8%). About 42.3% were males and the mean age was 22.3 ± 1.6 years. Table 1 shows participants’ demographic characteristics. The questionnaire was deemed reliable based on the result of Cronbach alpha coefficient (0.759) and its 95% CI (0.728−0.788).

Table 1 Participant demographic data (n = 504).

Age (Year), Mean ± SD
Age range	22.3 ± 1.6
20–29	
Gender, N (%)		
Male	213 (42.3)	
Female	291 (57.7)	
University, N (%)		
All Public Universities in Jordan	504 (100)	
Level of Education, N (%)		
3rd year	131 (26.0)	
4th year	63 (12.5)	
5th year	113 (22.4)	
6th year	197 (39.1)	
Know someone with cervical cancer, N (%)		
Yes	22 (4.4)	
No	482 (95.6)	
Nationality, N (%)		
Jordanian	454 (90.1)	
Not Jordanian	50 (9.9)	
Place of living, N (%)		
Urban	395 (78.4)	
Rural	109 (21.6)	
Family Income JD, N (%)		
<500	33 (6.5)	
501–1000	128 (25.4)	
1001–1499	111 (22.0)	
1500–2000	94 (18.7)	
>2000	138 (27.4)	
Notes.

N number

SD standard deviation

Knowledge assessment about CC

Knowledge was assessed using 14 questions with a total of 34 points related to disease diagnosis, risk factors, symptoms, and relation to HPV. In regards to knowledge about CC, the mean knowledge score for students was 21.4 ± 4.4. Table 2 shows the proportion of students who correctly answered questions related to CC and HPV. Most participants knew that CC is caused by an infection (n = 413, 81.9%) and that HPV is responsible for a wide array of diseases including CC (n = 455, 90.3%).

Table 2 Participant’s knowledge about cervical cancer (n = 504).

Question	Correct answer
N (%)	Wrong answer
N (%)	
Epidemiology of cervical cancer			
1. Is Cervical cancer the leading cause among gynecological cancer?1	306 (60.7)	198 (39.3)	
2. The cause of cervical cancer
Mean % score = 71.3% ± 30.8%	413 (81.9)	91 (18.1)	
3. Risk factors of cervical cancer			
- Multiple sexual partner	422 (83.7)	82 (16.3)	
-Infection with HPV	482 (95.6)	22 (4.4)	
-Early age of first coitus	245 (48.6)	259 (51.4)	
-Smoking	301 (59.7)	203 (40.3)	
-Family History of disease	324 (64.3)	180 (35.7)	
-Poor Hygiene	201 (39.9)	303 (60.1)	
-Old age (False)	366 (72.6)	138 (27.4)	
-Contraception	178 (35.3)	326 (64.7)	
-Nulliparity (False)	426 (84.5)	78 (15.5)	
Mean % score = 64.9% ± 21.0%			
4. Clinical features of cervical cancer			
-No symptom	179 (35.5)	325 (64.5)	
-Lower pelvic pain	200 (39.7)	304 (60.3)	
-Bleeding per vagina	406 (80.6)	98 (19.4)	
-Fever (False)*	441 (87.5)	63 (12.5)	
-Discharge per vagina	318 (63.1)	186 (36.9)	
Itching (False)*	353 (70.0)	151 (30.0)	
-Weight loss	264 (52.4)	240 (47.6)	
-Swelling of cervix (False)*	276 (54.8)	228 (45.2)	
-Anemia	202 (40.1)	302 (59.9)	
-Post coital bleeding	370 (73.4)	134 (26.6)	
Mean % score = 59.7% ± 16.5%			
Cervical cancer screening			
5. Time of screening for women aged 25–44 years	180 (35.7)	324 (64.3)	
6. Time of screening for women aged 45–60 years	133 (26.4)	371 (73.6)	
7. Is there a vaccine to protect from cervical cancer?1	386 (76.6	118 (23.4)	
8. Does the vaccine protect against all cervical cancer?2	332 (65.9)	172 (34.1)	
9. Girls who have been vaccinated will need to attend for cervical cancer screening1	359 (71.2)	145 (28.8)	
Mean % score = 55.2% ± 25.9%			
Knowledge about HPV			
10. Is HPV responsible for a wide range of diseases including cervical cancer?1	455 (90.3)	49 (9.7)	
11. Transmission of HPV	448 (88.9)	56 (11.1)	
12. The Technique available for HPV detection			
- Pap smear	376 (74.6)	128 (25.4)	
- Biopsy	169 (33.5)	335 (66.5)	
- PCR	218 (43.3)	286 (56.7)	
- Blood (False)*	390 (77.4)	114 (22.6)	
13. HPV subtypes 6 and 11 are commonly associated with Genital warts1	328 (65.1)	176 (34.9)	
14. HPV subtypes 16 and 18 are commonly associated with Cervical carcinoma1	355 (70.4)	149 (29.6)	
Mean % score = 67.9% ± 21.5%			
Overall % Knowledge score = 63.0% ± 12.9%			
Notes.

1 Yes.

2 No.

* Student get one point if the answer for this statement is false.

Concerning its epidemiology, 60.7% of students answered correctly that CC is a leading cause of gynecological cancer.

Student knowledge about the clinical features of CC and percentages of correct answers were 35.5% for no symptoms and 80.6% for bleeding of the vagina. The percentages of students who were aware that fever, itching, and swelling of the cervix were not among the clinical features of the disease were 87.5%, 70.0%, and 54.85% respectively. The mean percentage knowledge score of this part was 59.7% ± 16.5%.

Concerning knowledge about CC screening and vaccine, 26.4% of participants knew correctly that women aged 45–60 years should screen according to WHO once every 5 years. On the other hand, 76.6% were aware that there is a vaccine that protects from CC. The mean percentage knowledge score of this part was 55.2% ± 25.9%.

Most students (90.3%) knew that HPV is responsible for a wide array of diseases including CC. Most of them (88.9%) were aware that HPV is transmitted sexually. About two thirds of students were aware that HPV subtypes 6 and 11 are commonly associated with genital warts (65.1%) and HPV subtypes 16 and 18 are commonly associated with CC (70.4%). The mean percentage knowledge score of this domain was 67.9% ± 21.5%.

The association of socioeconomic factors with knowledge score was assessed using t-test and analysis of variance and there was a significant difference between the mean knowledge score for male students 20.6 ± 4.7 compared to female students 22.0 ± 4.0 (p-value = 0.001). The knowledge score was significantly associated with the year of study with the highest mean score among students in the sixth year with 23.6 ± 3.5 (p-value < 0.001) and the least mean score among third-year students 18.4 ± 4.2. There was no significant association between family income, place of living, or nationality and knowledge score (p-value >0.05).

Awareness about HPV vaccine and acceptance

Regarding student’s awareness about the HPV vaccine, only 40.5% knew about the availability of the HPV vaccine in Jordan, 71.4% were aware that the HPV vaccine should be given at an age between 11 and 29, and about half (54.0%) knew that it could be given to boys also. There were 21.0% that were familiar that girls or women do not need to be screened for HPV before getting vaccinated and that CC protection provided by the HPV vaccine is 70%. Less than one third of the students (30.8%) were aware that the HPV vaccine cannot be given to a woman already having HPV infection while 19.0% of students knew that three doses are required for protection in women. The mean awareness score of students about the HPV vaccine was 5.7 ± 2.8 with a range 0-13 which is classified as average awareness (Table 3).

Table 3 Participant’s awareness and acceptance of HPV vaccination (n = 504).

Question	Correct answer
N (%)	Wrong answer
N (%)	
1. Is the HPV vaccine available in Jordan?1
-Yes (2 points)
-No
-Don’t know	204 (40.5)
204 (40.5)
64 (12.7)
236 (46.8)	300 (59.5)	
2. Which age group HPV vaccine should be given?
-(0–10) Years
-(11–29) Years (2 points)
-(30–50) Years (1 point)
-(51) years and above	443 (87.9)*
48 (9.5)
360 (71.4)
83 (16.5)
13 (2.6)	51 (12.1)	
3. Can HP vaccine be given to boys?1
-Yes (2 points)
-No
-Don’t know	272 (54.0)
272 (54.0)
50 (9.9)
182 (36.1)	132 (46.0)	
4. Do girls/women need to be screened for HPV before getting vaccinated?2
-Yes
-No (2 points)
-Don’t know	109 (21.6)
237(47.0)
109 (21.6)
158 (31.4)	395 (78.4)	
5. Can HP vaccine be given to a woman already having HPV infection?2
-Yes
-No (2 points)
-Don’t know	155 (30.8)
141 (28.0)
155 (30.8)
208 (41.2)	349 (69.2)	
6. How many doses of HPV vaccine are required for protection in women?
-One
-Two
-Three (2 points)
-Four
-Don’t Know	96 (19.0)
49 (9.7)
70 (13.9)
96 (19.0)
8 (1.6)
281 (55.8)	408 (81.0)	
7. Cervical cancer protection provided by HPV vaccine is:
-100%
-90% (1 points)
-70% (2 point)
-50% (1 point)
-Don’t know	275 (54.6)*
21(4.2)
135(26.8)
106 (21.0)
34 (6.7)
208 (41.3)	263 (45.4)	
Overall % awareness score = 40.5% ± 19.8%	332 (65.9)	172 (34.1)	
Notes.

1 Yes.

2 No.

* Number of students who had a score of 1 or 2 points.

The awareness score about the HPV vaccine was significantly associated with the year level of study as there was a significant difference between the mean awareness score for students in the sixth year with 6.3 ± 2.8 (p-value < 0.001). On the other hand, the awareness score was not associated with all other demographic data (p-value > 0.05) as shown in Table 4.

Table 4 Knowledge and awareness about HPV vaccine scores and acceptance stratified by participants’ characteristics (n = 504).

		Knowledge score	Awareness score	Acceptance of the vaccine	
Variable	N (%)	Mean	SD	P-value	Mean	SD	P-value	Yes
N (%)	No
N (%)	Don’t know
N (%)	P-value	
Gender
Male
Female	
213 (42.3)
291 (57.7)	
20.6
22.0	
4.7
4.0	
<0.001*	
5.6
5.7	
2.7
2.8	
0.626	
138 (64.8)
194 (66.7)	
39 (18.3)
60 (20.6)	
36 (16.9)
37 (12.7)	
0.389	
Place of living
Urban
Rural	
395 (78.4)
109 (21.6)	
21.5
21.0	
4.2
5.0	
0.266	
5.7
5.7	
2.8
2.7	
0.959	
261 (66.1)
71 (65.1)	
80 (20.2)
19 (17.4)	
54 (13.7)
19 (17.4)	
0.551	
Nationality
Jordanian
Non-Jordanian	
454 (90.1)
50 (9.9)	
21.5
21.1	
4.4
4.6	
0.596	
5.6
6.0	
2.7
3.0	
0.431	
297 (65.4)
35 (70.0)	
94 (20.7)
5 (10.0)	
63 (13.9)
10 (20.0)	
0.140	
Know someone with Cervical Cancer
Yes
No	
22 (4.4)
482 (95.6)	
21.9
21.4	
5.2
4.4	
0.638	
6.2
5.6	
3.5
2.7	
0.373	
12 (54.6)
320 (66.4)	
3 (13.6)
96 (19.9)	
7 (31.8)
66 (13.7)	
0.060	
University
Yarmouk University
University of Jordan
Mutah University
The Jordan University of Science and Technology
Albalqa University
The Hashemite University	
168 (33.3)
81 (16.1)
32 (6.3)
126 (25.0)
29 (5.8)
68 (13.5)	
21.8
22.4
21.4
20.4
18.7
21.4	
4.3
4.4
4.5
4.4
4.0
4.0	
<0.001*	
5.9
5.7
5.2
5.4
5.2
6.1	
2.9
2.6
2.6
2.8
2.6
2.8	
0.272	
116 (69.0)
54 (66.7)
20 (62.5)
82 (65.1)
13 (44.8)
47 (69.1)	
30 (17.9)
15 (18.5)
7 (21.9)
26 (20.6)
8 (27.6)
13 (19.1)	
22 (13.1)
12 (14.8)
5 (15.6)
18 (14.3)
8 (27.6)
8 (11.8)	
0.644	
Year Level
Third year
Fourth year
Fifth year
Sixth year	
131 (26.0)
63 (12.5)
113 (22.4)
197 (39.1)	
18.3
20.0
22.1
23.6	
4.2
3.1
4.3
3.5	
<0.001*	
5.1
5.3
5.4
6.3	
2.6
2.6
2.8
2.8	
<0.001*	
90 (68.7)
39 (61.9)
63 (55.7)
140 (71.1)	
16 (12.2)
12 (19.0)
30 (26.6)
41 (20.8)	
25 (19.1)
12 (19.0)
20 (17.7)
16 (8.1)	
0.005*	
Family Income
<500
501–1000
1001–1499
1500–2000
>2000	
33 (6.5)
128 (25.4)
111 (22.0)
94 (18.7)
138 (27.4)	
22.0
21.0
21.1
21.7
21.8	
5.4
4.4
4.5
4.4
4.0	
0.502	
6.2
5.6
5.5
5.9
5.6	
3.0
2.9
2.8
2.7
1.7	
0.587	
19 (57.6)
87 (68.0)
77 (69.4)
62 (65.9)
87 (63.0)	
5 (15.2)
26 (20.3)
22 (19.8)
20 (21.3)
26 (18.8)	
9 (27.3)
15 (11.7)
12 (10.8)
12 (12.8)
25 (18.1)	
0.411	
Notes.

* P<0.05.

There were 332 students (65.9%) that accepted the idea that it is essential to introduce the HPV vaccine for schoolgirls in Jordan. The association of demographic data with the acceptance of introducing the HPV vaccine was assessed using chi-square analysis, and it was higher among students in the sixth year at 71.1% (p-value = 0.005). On the other hand, the acceptance of introducing the vaccine was not associated with all other demographic data nor with the knowledge score or awareness score (p-value > 0.05).

Among the obstacles that prevent the receipt of the vaccine or advice about taking HPV vaccination were high cost (53.8%) and inadequate information about the vaccine (62.5%) as shown in Fig. 1.

Figure 1 Obstacle preventing form receiving or advice taking HPV vaccination.

Figure 2 presents the sources of information about HPV vaccination which were reported by students as medical school teaching being the main source of information (87.7%) followed by internet sources (33.3%) and books (23.2%).

Figure 2 Sources of knowledge and information on HPV vaccination.

Discussion

In the current study, general CC and HPV knowledge was moderate which was similar to the result of a study done with medical, dental, and nursing students in South India (Shetty et al., 2019). Most of participants in the present study were aware that CC is caused by an infection and that the HPV infection can lead to CC. These results show adequate knowledge about CC epidemiology and are similar to other studies conducted with health care professionals (Sherman et al., 2020; Almazrou, Saddik & Jradi, 2019).

In our study, most medical students were able to recognize that infection and risky sexual practices are common risk factors for CC; these results were similar to other studies (Jolly et al., 2017; Melan et al., 2017; Goldsmith et al., 2007). However, some students had incorrect information that old age and nulliparity were among the risk factors of CC. In the current study, a high percentage of participants were unaware that lower pelvic pain and anemia are common clinical features of CC, and some of them wrongly thought that fever, itching, and swelling of the cervix were symptoms of CC. This highlights the need to increase the consciousness about CC among physicians who act as the main source of health information to their patients.

Less than half of students correctly reported PCR as a test used for the detection of HPV infection and knew the appropriate frequency of CC screening in women (i.e., women aged 25–44 years should be screened every three years and women aged 45–60 years should be screened every 5 years). This showed an inadequacy of knowledge which is important as a preventive measure for CC. In addition, the knowledge score was significantly associated with gender and year of study and this was consistent with other studies where the score was higher among female students (Hussain et al., 2014; Ngwenya & Huang, 2018) and higher level of study (Tesfaye et al., 2019).

The HPV vaccine offers a major breakthrough to limit the global burden of CC (Santhanes et al., 2018). Many studies have been conducted worldwide recently on the knowledge, attitude, beliefs, and awareness about the HPV vaccine (Sherman et al., 2020; Almazrou, Saddik & Jradi, 2019; Jradi & Bawazir, 2019; Maness et al., 2016; Kasymova, Harrison & Pascal, 2019). In the current study, only 40.5% of medical students were aware of the availability of the HPV vaccine in Jordan which protects from CC and about 20.0% were aware that there is no need to screen girls before getting vaccinated. Also, 70% were aware that CC protection is provided by the HPV vaccine. This could be due to the lack of awareness and guidance campaigns in Jordan to shed light on the importance of this vaccine, and it has not been introduced officially in the national vaccination program in Jordan. In addition, topics about cervical cancer and the HPV vaccine are not sufficiently addressed in the curricula of medical schools.

Less than one third of students were aware that the HPV vaccine cannot be given to a woman already having the HPV infection, and 19% of students knew that the appropriate number of doses of the HPV vaccine is three with an overall awareness score of 5.7 ± 2.8. This indicates an inadequate awareness about the HPV vaccine which was similar to a result from a study of university students in India (Gollu & Gore, 2021).

In the current study, about two thirds of students (65.9%) thought that it is important to introduce the vaccine in school girls in Jordan which indicates a favorable acceptance of using the vaccine in Jordan. Hoque (2016) indicated that most of the physicians in their study reported that they intended to prescribe the HPV vaccine to patients as they expected an important advantage from HPV vaccination.

In the present study, more than half of the students (62.5%) reported inadequate information about CC and the HPV vaccine as an obstacle preventing the receipt of the vaccine or advice about taking HPV vaccination. Our results were similar to a study on medical students in India (Pandey et al., 2012). Therefore, it is recommended that physicians should receive information about HPV from educational campaigns to improve their communication practices for recommending HPV vaccination (Hswen et al., 2017).

Strength and limitations of the study

The high response rate and the inclusion of all medical colleges in Jordan enhance the generalizability of our results. However, this is a cross sectional study and therefore causal relationships between variables cannot be established. We could also not detect the response rate as the questionnaire was distributed electronically. In addition, a systematic sampling method must be used to identify respondents from each level, but a convenient sampling method was used in the current research to reach the determined sample number. Our study is considered a starting point for future studies on this sensitive topic exploring the attitudes and barriers to vaccination among women in Jordan.

Conclusion

This study highlights insufficient knowledge about CC and its screening among medical students in Jordan. Despite the limited awareness about the HPV vaccine among the study’s participants, there is a favorable opinion towards the introduction of the vaccine in school girls in Jordan. More emphasis should be placed on the medical curriculum taught in undergraduate education. Suitable educational campaigns should be stratified at hospitals along with workshops and seminars which highlight the importance of CC screening in women and increase the awareness about HPV among physicians. Medical students who are the future health care providers can educate their patients, address their sensitive cultural concerns, and later increase the health seeking behavior in women in Jordan especially if they are properly aware of CC and hence its burden reduced.

Supplemental Information

Supplemental Information 1 Supplementary file

Click here for additional data file.

Supplemental Information 2 Questionnaire and consent form

Click here for additional data file.

Thanks to all the members of the medical students who participated in this study.

Additional Information and Declarations

Competing Interests

Author Contributions

Human Ethics

Data Availability

The authors declare there are no competing interests.

Mervat M. Alsous conceived and designed the experiments, performed the experiments, analyzed the data, prepared figures and/or tables, authored or reviewed drafts of the paper, and approved the final draft.

Ahlam Ali, Sayer Al-Azzam, Reema Karasneh and Haneen Amawi conceived and designed the experiments, performed the experiments, authored or reviewed drafts of the paper, and approved the final draft.

The following information was supplied relating to ethical approvals (i.e., approving body and any reference numbers):

The Institutional Review Board (IRB) of Jordan University of Science and Technology (JUST) and King Abdulla University Hospital (KAUH), Irbid, Jordan approved this research (13/128/2019).

The following information was supplied regarding data availability:

The data are available in the Supplemental Files.

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
