# Peer review of "Knowledge about cervical cancer and awareness about human papillomavirus vaccination among medical students in Jordan"

_PeerJ, doi:10.7717/peerj.11611_

## Round 0.1 · original submission · Major Revisions

Your manuscript was considered interesting and valuable, and one of the reviewers actually commended you on your extensive data collection. However, there were a few of issues raised by the reviewers that need to be addressed. The reviewers want you to provide more detail in several sections of your manuscript, such as the introduction (regarding the different oncogenic HPV strains among others), in the methods and data analysis sections so your analyses can be independently reproduced, in Table 3, and in the discussion section.

Please, submit a detailed rebuttal which shows where and how you have taken all comments and suggestions into consideration. If you do not agree with some of the reviewers’ comments or suggestions, please explain why. Your rebuttal will be critical in making a final decision on your manuscript. Please, note also that your revised version may enter a new round of review by the same or by different reviewers. Therefore, I cannot guarantee that your manuscript will eventually be accepted.

Reviewer 1 ·

Basic reporting

The language of the article is clear, understandable and technically correct text. The article conform to professional standards of courtesy and expression.
The article, include introduction and background to demonstrate how the work fits into the broader field of knowledge. Suggested additions should be made in the introduction.
The structure of the article on form to an acceptable format of ‘standard sections’ . Tables is relevant to the content of the article. All appropriate raw data have been made available in accordance with your Data Sharing policy.
The submission is include all results relevant to the hypothesis.

Experimental design

The article is original research within Aims and Scope of the journal.
Research question well defined, relevant & meaningful. It have been stated how research fills an identified knowledge gap.
This investigation was performed to a technical & ethical standard.
Methods are defined but not sufficient. The method of the study (scoring the qestions) may not be used by other investigator.

Validity of the findings

Impact and novelty not assessed.
The results of the study are widely known and expected results. The definition of the results is important but not a novelty.
All underlying data have been provided.
Conclusions are stated, linked to original research question.

Additional comments

I commend the authors for their extensive data set. Student information level section in the survey does not include the 3th year (4,5,6) option. Can be added for other researchers.
In addition, the manuscript is "Knowledge about cervical cancer and awareness about human papillomavirus vaccination among medical students in Jordan" is addressing a current issue. The weakness strengths of the article is related to generally the method section (as I have noted below).
- It should be stated in the article that random sample selection was made to reach the determined sample (actually my suggestion:In order to reach the determined sample number, the systematic sampling method must be used to identify respondents from each level).
- It is stated that the scoring of the knowledge and awareness questions created according to the literature information was made by the authors. In the article, no information was found regarding the content validity (expert opinion and cronbach's alpha value) and reliability of these questions. If it is, it should be added to the article. This is important for the accuracy of the measurement of the questions.
- The total number of students in all faculties should be given.
- in Statistical Analysis section, line 180: "Chi square was used to analyze" statement is stated. However, this analysis was not seen in the article.
- Also, "Student's t-test was used to compare the means between two groups" was made. This situation is expressed in the conclusion section and findings are given. However, it was determined that it was not shown as a table. These analysis results should also be shown in a table Showing these findings in the Table may increase the attention of the researchers.
Recommendations outside the method section:
-Your listing needs a little more detail. To provide more justification for your study, examples of studies on medical students' knowledge and awareness of HPV can be given after 118 lines. I suggest you improve the explanation on whether it is being done for the first time with medical students in Jordan. Also, the first paragraph in the discussion section contains information. I recommend it to be included in the introduction (between 243-256 lines).
- Sources compared in the discussion should be similar studies. For example, Discussion section, line 252-253:"In the current study, general CC and HPV knowledge was moderate which was similar to the result of a study done on nurses in Thailand [18]". The results of the work done on students should be compared instead of the 18th source used in the discussion. In addition, 30 referances are a study done only for female students.

Reviewer 2 ·

Basic reporting

The English language should be check on by a professional translator.
The Introduction of the manuscript has to be modified. I suggest that author
to provide more data about other oncogenic HPV types in the world that are connected with CC and about types of vaccines against HPV, at lines 92, 96- 97 .
Is there any type of vaccine against HPV in Jordan? Is the vaccination against HPV mandatory or just a recommendation ? Is it only for girs? What about vaccination of boys?
The manuscript has the Introduction, the Objective, Methods, Results, Discussion and Conclusion.
Table 3. Check the sum of percentage on the 4th and the 5th question. The sum is not 100%.
Both figures shuld be modified to be more clear. For example: 28% above the column, and below the line of the graph, the legend shoul be placed: Percentage (%)

No comment

Experimental design

No comment

Validity of the findings

No comment

Additional comments

Very Respected author,
In order to improve Your manuscript I suggest to modifie the Introduction and to add more data about types of oncogenic HPV, more about type of vaccines aginst HPV, and more data about HPV screening and about Pap test.

Reviewer 3 ·

Basic reporting

yeah English language is good, Literature references has been cited well. tables and figures are also good

Experimental design

Yeah experimental design is good, Method has been explained properly.

Validity of the findings

i checked on PubMed about previously published article related to Knowledge about cervical cancer in Jordan peoples. There are some publication about related topic but no reported Knowledge about cervical cancer among medical students in Jordan. therefore i think this study can be considered for publication.
Finding are good but i suggest them to discuss the results with more detail.
Please explain knowledge and vaccine acceptance gaps among rural and urban peoples, also explain the difference of knowledge and vaccine acceptance gaps among Level of Education

Additional comments

This is good study but it need to revised before acceptance
Please add results about Knowledge about cervical cancer and vaccine acceptance gaps among medical students with rural or urban background.
also results about knowledge and vaccine acceptance gaps among Level of Education, nationality, family income etc.
Please why medical students do not know about availability of HPV vaccine in Jordan. Explain the reason and discuss it accordingly.
Pleaser discuss Table 3: Participant’s awareness and acceptance of HPV vaccination (n=504) results in detail.

---

## Round 0.2 · accepted · Accept

Thank you for thoroughly addressing the reviewers' comments. Your manuscript is significantly improved as a result.

Reviewer 2 ·

Basic reporting

Authors significantly improved their manuscript. I do not have any negative comments.

Experimental design

No comment

Validity of the findings

No comment

Additional comments

No comment

Reviewer 3 ·

Basic reporting

Revised version has been improved according to suggestions

Experimental design

now Experimental design has been explain accordingly

Validity of the findings

Findings are interesting for the readers